# Deletion of 3p13-14 locus spanning *FOXP1* to *SHQ1* cooperates with *PTEN* loss in prostate oncogenesis

Haley Hieronymus[1], Phillip J. Iaquinta[1], John Wongvipat[1], Anuradha Gopalan[2], Rajmohan Murali[2], Ninghui Mao[1], Brett S. Carver[1,3] & Charles L. Sawyers[4]

A multigenic locus at 3p13-14, spanning *FOXP1* to *SHQ1*, is commonly deleted in prostate cancer and lost broadly in a range of cancers but has unknown significance to oncogenesis or prognosis. Here, we report that *FOXP1-SHQ1* deletion cooperates with *PTEN* loss to accelerate prostate oncogenesis and that loss of component genes correlates with prostate, breast, and head and neck cancer recurrence. We demonstrate that *Foxp1-Shq1* deletion accelerates prostate tumorigenesis in mice in combination with *Pten* loss, consistent with the association of *FOXP1-SHQ1* and *PTEN* loss observed in human cancers. Tumors with combined *Foxp1-Shq1* and *Pten* deletion show increased proliferation and anaplastic dedifferentiation, as well as mTORC1 hyperactivation with reduced Akt phosphorylation. *Foxp1-Shq1* deletion restores expression of AR target genes repressed in tumors with *Pten* loss, circumventing PI3K-mediated repression of the androgen axis. Moreover, *FOXP1-SHQ1* deletion has prognostic relevance, with cancer recurrence associated with combined loss of *PTEN* and *FOXP1-SHQ1* genes.

[1] Human Oncology and Pathogenesis Program, Memorial Sloan Kettering Cancer Center, 1275 York Ave, New York, NY 10065, USA. [2] Department of Pathology, Memorial Sloan Kettering Cancer Center, 1275 York Ave, New York, NY 10065, USA. [3] Department of Urology, Memorial Sloan Kettering Cancer Center, 1275 York Ave, New York, NY 10065, USA. [4] Howard Hughes Medical Institute, Chevy Chase, MD 20815, USA. Correspondence and requests for materials should be addressed to C.L.S. (email: sawyersc@mskcc.org)

With the advent of comprehensive genomic characterization of many cancers, understanding the functional relationships between common genomic alterations in cancer has become imperative. In addition to individual driver or passenger genes, many such alterations are multigenic and pose questions about the combined contributions of multiple genes to oncogenesis[1, 2]. Recent work, for example, has found that the 17p deletion harboring *TP53* increases leukemia development beyond that from *TP53* loss alone and that multiple genes within the deletion cooperate with *TP53* loss[2]. Such cooperativity within multigenic deletions may contribute to oncogenesis on a wider basis.

Prostate cancer, the most common malignancy in men, shows a significant multigenic deletion at 3p13-14 that spans *FOXP1*, *EIF4E3*, *GPR27*, *PROK2*, *RYBP*, and *SHQ1*[3, 4]. It is unknown whether this region has a tumor suppressive function, but more than 12% of primary prostate tumors show copy number loss of the 3p13-14 region from *FOXP1* to *SHQ1* (termed *FOXP1-SHQ1* deletion) and 15% show copy number loss of the individual *FOXP1* or *SHQ1* genes that bound the deletion. Deletion of the *FOXP1-SHQ1* region is seen in a wide variety of cancers, ranging from 27% of primary breast cancer[5] and stomach cancer[6] to 70% or more in renal clear cell[7] and head and neck[8] cancers, often within larger deletions.

Although none of the genes encoded within the 3p13-14 *FOXP1-SHQ1* locus are known to be canonical tumor suppressors, several have potential tumor suppressive roles. SHQ1, an RNP assembly factor required for ribosome and telomerase function, acts as a co-factor to dyskeratosis congenita protein DKC1 mutated in resulting malignancies[9]. EIF4E3 is a cap-binding translation initiation factor that competes with EIF4E1[10] and suppresses translation of its growth-promoting targets[10, 11]. RYBP, a component of the RING1B-containing polycomb repressive complex and others, is required for histone modification[12, 13] and resultant cellular differentiation[14]. The forkhead transcription factor FOXP1 is a potential tumor suppressor[15] that may regulate the androgen receptor (AR)[16] and estrogen receptor[17] signaling axes in prostate and breast cancer.

Of the 3p13-14 deletion genes, recurrent somatic mutations have been reported in *FOXP1*, *RYBP*, and *SHQ1*[18]. *FOXP1* has the highest number of somatic point mutations in prostate cancer, including H515R/Y and the adjacent L519del in the forkhead domain, which are also seen in other cancers[18]. *SHQ1*, though infrequently mutated, has been found to have a A228T mutation found in other cancer types, as well as additional mutations in the SHQ1-dyskerin interaction domain[4]. Thus, deletion of the 3p13-14 *FOXP1-SHQ1* region may confer loss of multiple tumor suppressor functions.

Here, we provide conclusive evidence that the 3p13-14 *FOXP1-SHQ1* deletion is a driver event in prostate oncogenesis through interaction with other genetic alterations, most notably phosphoinositide 3-kinase (PI3K) pathway activation by *PTEN* loss. Furthermore, human tumors with *FOXP1-SHQ1* deletion are enriched in *PTEN* deletion. *FOXP1-SHQ1* loss relieves *PTEN* loss-mediated repression of the critical AR signaling pathway, in addition to hyperactivating the mTORC pathway. Clinically, we find that combined *FOXP1* and *PTEN* loss in human prostate, breast cancer, and head and neck cancer correlates with increased cancer recurrence, suggesting that combined deletion has functional and prognostic significance in cancer.

## Results

### 3p13-14 and *PTEN* deletions are co-enriched in cancer.
To explore the genetic interactions between the 3p13-14 *FOXP1-SHQ1* deletion and other major cancer alterations, we assessed whether prostate cancers with this deletion showed significant co-enrichment of other focal copy number alterations. We find that *FOXP1-SHQ1* deletion co-occurs more often than expected with loss of *PTEN*, a common tumor suppressor of the PI3K pathway, in primary prostate adenocarcinoma (Memorial Sloan Kettering cohort[3], $P = 0.008$, Fisher's exact test, Supplementary Table 1, Fig. 1a). *PTEN* loss occurs in 40% of prostate tumors that show *FOXP1-SHQ1* deletion, but in only 13% overall. *PTEN* loss is also positively associated with *FOXP1* copy number loss ($P = 0.001$, Fisher's exact test) or *SHQ1* copy number loss ($P = 0.007$, Fisher's exact test), when these are assessed individually (Fig. 1a). These associations are validated by the TCGA primary prostate adenocarcinoma cohort (Fig. 1b, Supplementary Table 1)[4]. Moreover, *PIK3CB* mutation and amplification is significantly enriched in tumors with *FOXP1-SHQ1* deletion ($P = 0.002$, Fisher's exact test) in the TCGA prostate cancer cohort, which has available exome sequencing data (Supplementary Table 1). Interestingly, the association between *PI3KB* mutation and *FOXP1-SHQ1* loss primarily occurs in the context of *PTEN* loss ($P = 0.004$ in the subset with *PTEN* loss, NS in the subset without *PTEN* loss, Fisher's exact test). Since *PTEN* deleted tumors are likely dependent on *PIK3CB* due to feedback inhibition of *PIK3CA*[4], activating *PIK3CB* mutations such as the observed E552K mutation[19] may be a mechanism of enhancing the oncogenicity associated with *PTEN* deletion. Neuroendocrine prostate cancer[20] also shows co-enrichment of *PTEN* loss and *FOXP1-SHQ1*deletion ($P = 0.004$, Fisher's exact test, Supplementary Table 1). Together, these findings suggest that activating PI3K pathway alterations are associated with 3p13-14 *FOXP1-SHQ1* deletion in prostate cancer.

While we previously found that *FOXP1-SHQ1* deletion is co-enriched with *ERG* fusion in addition to *PTEN* loss in human prostate cancer[3], *PTEN* loss is not co-enriched with *ERG* fusion within the *SHQ1-FOXP1* deleted population ($P = 0.24$, Fisher's exact test, prostate TCGA cohort[4]). *ERG* fusion and *PTEN* loss are known to be co-enriched in unselected patients and functionally cooperate in oncogenesis, but we find that their statistical co-enrichment is only seen in human prostate cancer lacking *FOXP1-SHQ1* deletion ($P < 0.001$, Fisher's exact test, prostate TCGA cohort[4]). Conversely, it has been previously reported that *FOXP1* deletion is only co-enriched with *PTEN* deletion in *ERG* fusion-negative tumors as ascertained by immunohistochemistry (IHC) and fluorescence in situ hybridization (FISH)[21], though we do not see this association in other genomic cohorts (prostate TCGA[4], MSKCC[3], $P > 0.1$). Prostate cancer with 3p13-14 *FOXP1-SHQ1* loss nonetheless shows statistically independent enrichment of *PTEN* loss and *ERG* fusion, which themselves only show co-enrichment with each other in cancers without *FOXP1-SHQ1* loss.

Since *PTEN* loss occurs in many cancer types, we next investigated the broader significance of combined *PTEN* and 3p13-14 *FOXP1-SHQ1* loss in cancer by determining whether *PTEN* and *FOXP1-SHQ1* copy number loss is co-enriched across other cancers. Of 12 published TCGA cohorts showing *FOXP1-SHQ1* deletion and *PTEN* deletion individually in at least 2% of tumors, 7 cancer types showed significantly enriched co-deletion of the two regions (Fig. 1c, Supplementary Table 2). One additional TCGA cohort, bladder urothelial carcinoma, showed a significant association between *FOXP1-SHQ1* deletion and *PIK3CA* mutation or amplification (Supplementary Table 2). Combined *PTEN* and *FOXP1-SHQ1* loss therefore broadly occurs across many cancer types.

### Foxp1-Shq1 and Pten loss cooperate in prostate oncogenesis.
To address the role of *FOXP1-SHQ1* deletion in oncogenesis and

its potential cooperativity with *PTEN* loss, we conditionally deleted the syntenic *Foxp1-Shq1* region in transgenic mice (Supplementary Fig. 1a–d; Supplementary Methods). In this *Foxp1-Shq1^flox* mouse line, *Foxp1* (exons 11 and 12) and *Shq1* (exon 2 with frameshift) are each flanked by loxP sites on the same chromosome for a total of four sequential loxP sites. Cre introduction results in conditional deletion of *Foxp1* and *Shq1* individually through recombination at their respective loxP pairs or deletion of the larger region from *Shq1* (exon 2) to *Foxp1* (exon 12) through the outer loxP sites (Supplementary Fig. 1d). Given the lower efficiency of whole locus recombination, this combinatorial design of loxP sites allowed us to test whether deletion of the multigenic region confers selective advantage over deletion of either or both of the two genes at the edges of deletion. Prostate-specific deletion was generated by introducing Cre expressed by the prostate-specific probasin promoter (*Pb-Cre4*). Combination with *Pten* loss was investigated by crossing with *Pten* conditional knockout (*Pten^flox/flox*) mice[22] to generate double transgenic *Foxp1-Shq1^f/f;Pten^f/f* mice. *Foxp1-Shq1^f/f* mice show heterogeneous deletion of the full *Foxp1-Shq1* locus as well as of the individual *Foxp1* and *Shq1* genes (Supplementary Fig. 1e), likely due to a mix of recombination events in different cells and sister chromosomes.

To test whether *Foxp1-Shq1* deletion cooperates functionally with *Pten* loss, we compared prostate phenotypes in mice with both *Foxp1-Shq1* and *Pten* deletion (*Foxp1-Shq1^f/f;Pten^f/f*) to those with only *Pten* deletion (*Pten^f/f*) or *Foxp1-Shq1* deletion (*Foxp1-Shq1^f/f*). *Foxp1-Shq1^f/f;Pten^f/f* mice showed accelerated oncogenesis with highly anaplastic histopathology compared to *Pten^f/f* mice, while *Foxp1-Shq1^f/f* mice showed no obvious prostate cancer phenotype (Figs. 2a, b; Supplementary Fig. 2). In the setting of *Pten* loss, eight of nine *Foxp1-Shq1^f/f;Pten^f/f* tumors tested by genomic PCR showed recombination of the full *Foxp1-Shq1* locus, with one showing exclusive *Foxp1-Shq1* locus deletion and seven showing a mix of full *Foxp1-Shq1* locus deletion and individual *Foxp1* and *Shq1* deletion (Supplementary Fig. 1e). Furthermore, the one prostate without full locus deletion had robust deletion of *Foxp1* and *Shq1*. The ratio of *Foxp1-Shq1* locus recombination to *Foxp1*-specific recombination averaged 0.38 by quantitative PCR of genomic DNA (12-month-old mouse prostate tumors), though this efficiency of locus recombination was variable between tumors as indicated by the standard deviation of 0.13. Genes within the *Foxp1-Shq1* interval, including *Shq1*, *Foxp1*, *Eif4e3*, *Prok2*, and *Rybp*, also show decreased RNA expression in *Foxp1-Shq1^f/f;Pten^f/f* prostates compared to *Pten^f/f* alone (by quantitative PCR, Fig. 2c; by RNA-seq, Fig. 2d), similar to human prostate cancers with the deletion. *Prok2* RNA levels show a greater decline than that of other locus genes (Fig. 2c), possibly because Prok2 exhibits positive feedback activation of its own transcription[23] and heterogeneous deletion of this secreted cytokine may result in greater reduction in its levels in adjacent cells with incomplete locus deletion. Since the human 3p13-14 locus more commonly exhibits mono-allelic loss than bi-allelic deletion in prostate cancer and its component genes show an average 1.5-fold reduction in RNA expression upon mono-allelic loss[3], the *Foxp1-Shq1^f/f;Pten^f/f* mouse model exhibits similar level of 3p13-14 gene expression change to that of human disease. In sum, both DNA and RNA analysis of prostate tissue confirmed deletion of the entire *Foxp1-Shq1* locus similar to that seen in human prostate cancer.

Combined loss of *Foxp1-Shq1* and *Pten* resulted in significant acceleration of cancer development compared to *Pten* loss alone in mice. All *Foxp1-Shq1^f/f;Pten^f/f* mice developed intraductal carcinoma by 6 months compared to 43% of *Pten^f/f* mice, and 60% developed invasive carcinoma[24] by 9 months of age compared to no *Pten^f/f* mice at 9 months and less than 30% of

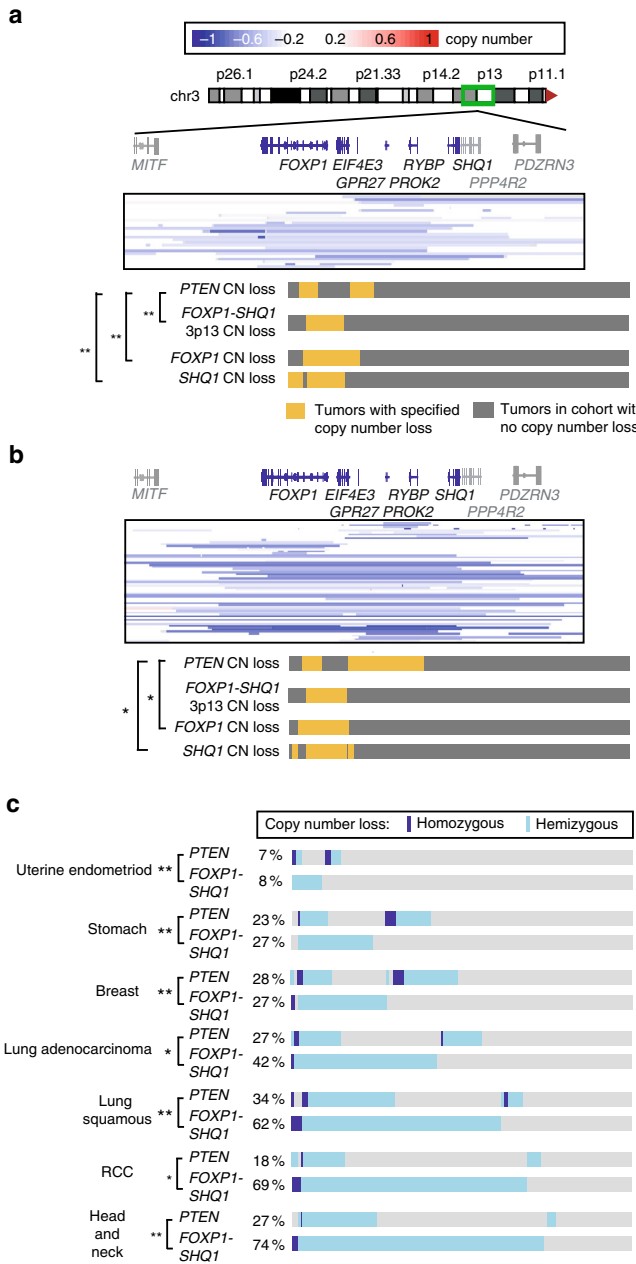

**Fig. 1** 3p13-14 *FOXP1-SHQ1* loss is associated with *PTEN* loss in human cancer. **a** In the MSKCC primary prostate cancer cohort (n = 146), *PTEN* copy number loss is enriched in tumors with copy number loss of *FOXP1*, *SHQ1*, and the 3p13-14 locus spanning *SHQ1* to *FOXP1* (homozygous or hemizygous). The boxed heat map shows copy number loss per tumor in samples with copy number loss at the 3p13-14 locus. The bar graph below shows the hemizygous or homozygous copy number loss (yellow) or absence of loss (gray) of the specified gene per tumor across the whole cohort. Significance of association, Fisher's exact test, *$P < 0.05$; **$P < 0.01$. **b** In the TCGA prostate adenocarcinoma cohort (n = 330), *PTEN* copy number loss is enriched in tumors with copy number loss of *FOXP1* and *SHQ1*. The boxed heat map and bar graph show copy number loss as in **a**. **c** Multiple cancer types show co-enrichment of *PTEN* copy number loss with *FOXP1-SHQ1* copy number loss. The bar graphs show the hemizygous copy number loss (light blue), homozygous copy number loss (dark blue), or absence of loss (gray) per tumor across the TCGA cohorts. The percentage of each cohort with specified copy number loss is shown. Significance of association, Fisher's exact test, *$P < 0.05$; **$P < 0.01$

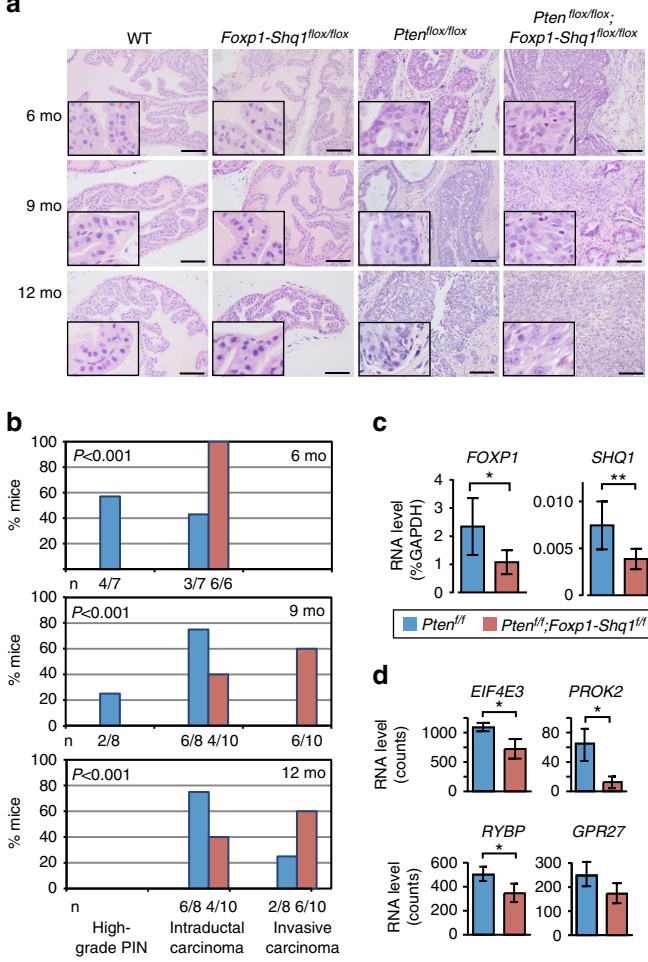

**Fig. 2** *Foxp1-Shq1* loss cooperates with *Pten* loss in prostate tumorigenesis. **a** Comparison of prostate histology (H&E) in wild-type, *Foxp1-Shq1^flox/flox^*, *Pten^flox/flox^*, and *Foxp1-Shq1^flox/flox^;Pten^flox/flox^* mice with *Pb-Cre* at 6, 9, and 12 months. Scale bars, 50 μm. Insets show 4× magnification of a portion of the full panel. **b** Histogram of histology in *Pten^flox/flox^* (blue; total $n = 23$; 6 mo, $n = 7$; 9 mo, $n = 8$; 12 mo, $n = 8$) and *Foxp1-Shq1^flox/flox^;Pten^flox/flox^* (red; total $n = 28$; 6 mo, $n = 8$; 9 mo, $n = 10$; 12 mo, $n = 10$) mouse prostates at 6, 9, and 12 months. The number, $n$, mice is listed as a fraction of the total $n$ per genotype per time point below each bar. P, Chi-square significance value. **c** Quantification of *Foxp1* and *Shq1* RNA level by quantitative PCR normalized to GAPDH in *Pten^flox/flox^* and *Foxp1-Shq1^flox/flox^;Pten^flox/flox^* 12-month mouse prostates. Error bars, SD. Significance of association, *t*-test, *$P < 0.05$; **$P < 0.01$. **d** Quantification of the RNA level of *Eif4e3*, *Gpr27*, *Prok2*, and *Rybp* genes intergenic to *Foxp1* and *Shq1*, by RNA sequencing in *Pten^flox/flox^* and *Foxp1-Shq1^flox/flox^;Pten^flox/flox^* 12-month mouse prostates. Normalized counts are shown. Significance of association, *t*-test, *$P < 0.05$. Error bars, SD

knockout mice (embryonic lethal, $P < 0.001$, $\chi^2$ test, $n = 18$ wild-type, 30 heterozygous knockout, 0 homozygous knockout live births) and in yeast[9]. This indicates that *SHQ1* loss alone in this system is insufficient to cause the observed cooperative oncogenesis, despite reports that orthotopic *SHQ1* shRNA-based knockdown is sufficient to increase metastasis in a *Trp53^+/loxP^*;Pten^loxP/loxP^ background[25].

Tumors with combined loss of *Foxp1-Shq1* and *Pten* show altered pathology with anaplastic morphologic features compared to *Pten^f/f^* tumors alone (Fig. 2a). Histologic features of the *Foxp1-Shq1^f/f^;Pten^f/f^* tumors ranged from invasive carcinoma consisting of a well-differentiated glandular component transitioning to a poorly differentiated epithelial component with sheet-like architecture at 9 months (Fig. 2a). At 12 months, the tumors are composed of a predominantly poorly differentiated, sarcomatoid carcinoma (expressing both epithelial and mesenchymal markers by IHC) with marked nuclear pleomorphism and a high mitotic rate (Fig. 2a; Supplementary Figs. 3, 4). In *Foxp1-Shq1^f/f^;Pten^f/f^* murine tumors, cellular proliferation assayed by Ki67 IHC was significantly increased compared to *Pten^f/f^* tumors, which is itself higher than the low proliferation rates seen in *Foxp1-Shq1^f/f^* and wild-type prostates (Supplementary Fig. 3). *Foxp1-Shq1^f/f^;Pten^f/f^* tumors nonetheless retain expression of luminal markers such as CK8/18 as seen in human adenocarcinoma (Supplementary Fig. 4). Expression signatures of differentiation were repressed (FDR < 0.01, gene set enrichment analysis (GSEA)) in *Foxp1-Shq1^f/f^;Pten^f/f^* prostate tumors, which exhibit distinct gene expression patterns from *Pten^f/f^* tumors by unsupervised RNA-seq clustering as well as repression of *Foxp* factor-derived expression signatures when compared to *Pten^f/f^* tumors (Supplementary Fig. 5). These results indicate that *Foxp1-Shq1* deletion cooperates with *Pten* loss in murine models to promote prostate cancer.

***Foxp1-Shq1* loss hyperactivates mTORC pathway upon *Pten* loss**. Next, we examined changes in PI3K and AR signaling and other oncogenic pathways upon cooperative *Foxp1-Shq1* and *Pten* deletion. *Foxp1-Shq1^f/f^;Pten^f/f^* murine tumors show elevated S6 phosphorylation by IHC compared to *Pten^f/f^* tumors by 12 months but reduced Akt phosphorylation, suggesting a high level of mTORC1 activation with feedback inhibition of PI3K[26, 27] (Fig. 3a). In support of this conclusion, expression signatures from *Foxp1-Shq1^f/f^;Pten^f/f^* tumors were inversely associated with signatures of TORC1 inhibition, compared to *Pten^f/f^* tumors (Supplementary Fig. 5d). FOXP1 has been shown to repress both ERK and MEK signaling[28] and S6 activation[29] in CD8+ T cells, thereby maintaining their quiescence, raising the possibility that *FOXP1* loss could result in an increase in mTORC1 activity through modulation of these pathways. Consistent with MEK involvement, genes repressed by MEK activation are under-represented in prostate tumors from *Foxp1-Shq1^f/f^;Pten^f/f^* mice relative to *Pten^f/f^* mice at 12 months, as assessed by GSEA of RNA-seq expression data ($P = 0.004$, FDR = 0.05, GSEA, Supplementary Fig. 5e). Moreover, higher ERK phosphorylation is seen in *Foxp1-Shq1^f/f^;Pten^f/f^* mice relative to *Pten^f/f^* mice and *Foxp1-Shq1^f/f^* mice at 12 months by phospho-ERK IHC (Supplementary Fig. 5f, g), suggesting synergistic MEK activation upon combined deletion.

***Foxp1-Shq1* loss restores AR signaling repressed by *Pten* loss**. PI3K pathway activation via *PTEN* loss is known to reduce AR protein levels and signaling[30], and AR signaling in murine and human tumors lacking PTEN is associated with reduced expression of AR-regulated luminal epithelial genes[31]. As expected, we observed decreased AR protein level by IHC in *Foxp1-Shq1^f/f^*;

*Pten^f/f^* mice by 12 months (Fig. 2b, Supplementary Fig. 2). Notably, *Foxp1-Shq1* deletion did not cooperate with *Tmprss2-Erg* fusion in transgenic *Foxp1-Shq1^f/f^;R26^ERG^* mice ($n = 9$ per time point at 6, 9, 12, and 18 months), which we created to test the functional cooperativity between statistically co-enriched *FOXP1-SHQ1* deletion and *TMPRSS2-ERG* fusion[3]. Moreover, *Shq1^f/f^;Pten^f/f^* mice generated here did not show accelerated prostate oncogenesis ($n = 9$ per time point at 6, 9, 12, and 18 months), possibly due to the lethality of complete *SHQ1* loss in germline

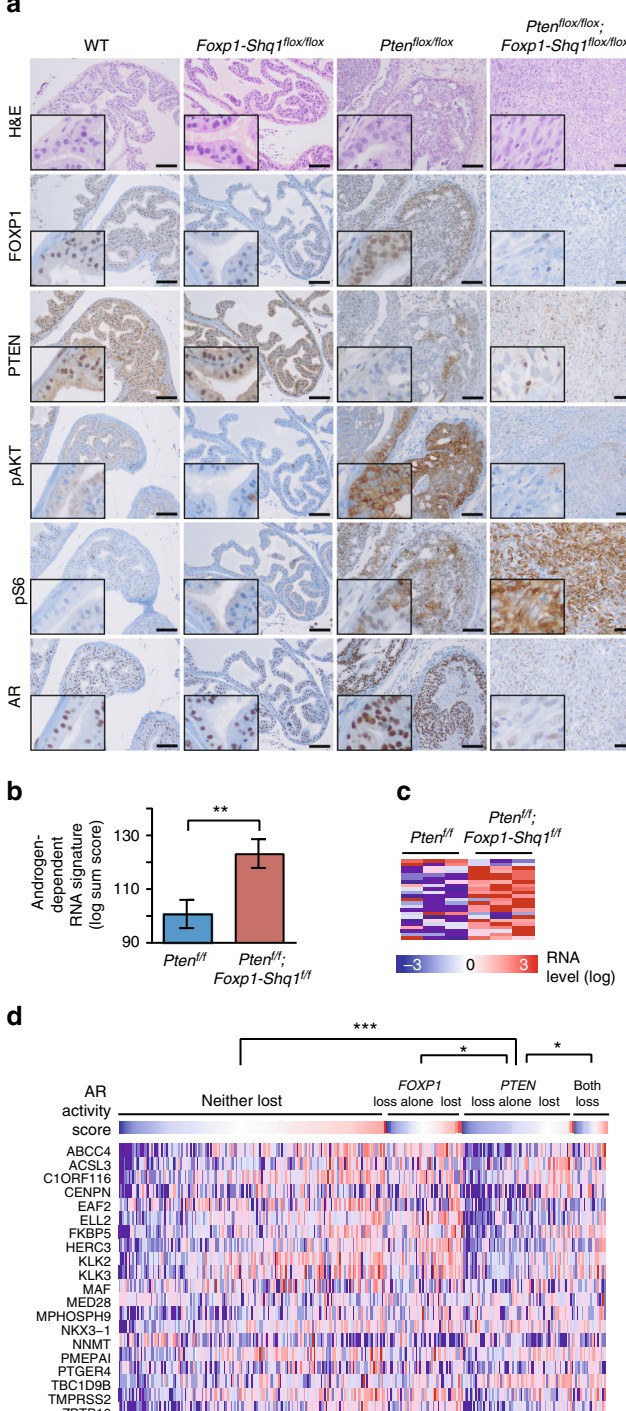

**Fig. 3** PI3K and AR pathways are altered upon cooperative loss of *Pten* and the *Foxp1-Shq1* locus, resulting in restoration of androgen-responsive gene expression. **a** Comparison of prostate histology (H&E) and FOXP1, PTEN, phospho-AKT (Ser473), phospho-S6 (Ser235/236), and AR levels by IHC in wild-type, *Foxp1-Shq1^{flox/flox}*, *Pten^{flox/flox}*, and *Foxp1-Shq1^{flox/flox};Pten^{flox/flox}* mice at 12 months. Main panel, scale bars, 50 μm. Inset, 4× magnification of main panel. **b** Sum score and **c** heat map of murine androgen-induced signature[30] in *Foxp1-Shq1^{flox/flox};Pten^{flox/flox}* (red) and *Pten^{flox/flox}* (blue) mouse prostate at 12 months by RNA-seq (n = 3 biological replicates per genotype). t-test, **P < 0.01. Error bars, SD. **d** Heat map of androgen-regulated signature and score[4, 50] in primary human prostate adenocarcinoma (TCGA cohort) grouped by *PTEN* loss and *FOXP1* loss status. All comparisons tested. t-test, *P < 0.05; **P < 0.01; ***P < 0.001

*Pten^{f/f}* tumors compared to *Foxp1-Shq1^{f/f}* and wild-type murine prostate at 12 months (Fig. 3a), whereas AR RNA levels are unchanged (Supplementary Fig. 6a). However, while *Pten^{f/f}* prostate tumors showed decreased RNA expression of androgen-dependent genes[30], *Foxp1-Shq1^{f/f};Pten^{f/f}* tumors showed a restoration of androgen-dependent gene expression signatures compared to *Pten^{f/f}* tumors at 12 months by RNA-seq (Fig. 3b, c; Supplementary Fig. 6b). Androgen-dependent gene expression is also decreased in human prostate cancers with *PTEN* loss alone (compared to tumors without *PTEN* loss) but is partially restored in tumors with combined *PTEN* and *FOXP1* loss (Fig. 3d, Supplementary Fig. 6c). In human prostate tumors, *FOXP1* loss added to *PTEN* loss correlates with significantly higher levels of androgen-responsive signature gene expression than the repressed levels caused by *PTEN* loss alone, but shows no significant difference in androgen-responsive signature expression compared to tumors without loss or with only *FOXP1* loss alone (Fig. 3d, Supplementary Fig. 6c). These results suggest that *FOXP1-SHQ1* deletion partially restores androgen-dependent gene expression in the context of *PTEN* loss. This androgen signaling rescue could potentially occur through loss of FOXP1, which is known to bind a portion of the AR cistrome and partially inhibit androgen-regulated gene expression[16]. The observed de-repression of canonical androgen-regulated genes may occur through de-repression of AR, which is expressed albeit at lower levels (Supplementary Fig. 6a, d); this would thereby increase the active or non-repressed pool of AR. *FOXP1-SHQ1* deletion may therefore be selected in *PTEN*-null prostate cancers as a mechanism to modulate the inhibitory effects of PI3K pathway activation on androgen signaling.

**Combined 3p13/PTEN loss is associated with cancer recurrence.** Since *Foxp1-Shq1* deletion and *Pten* loss cooperate together to promote murine prostate oncogenesis, we next asked whether combined loss of *PTEN* and *FOXP1-SHQ1* locus genes were associated with worse outcome in human cancer. While prostate cancer recurrence was not statistically significantly associated with combined genomic loss of *PTEN* and *FOXP1-SHQ1*, inclusion of Foxp1 expression loss revealed a prognostic association with recurrence. Specifically, combined *FOXP1* and *PTEN* loss (defined as copy number or expression loss) is associated with prostate cancer recurrence in multiple cohorts (Fig. 4). In the

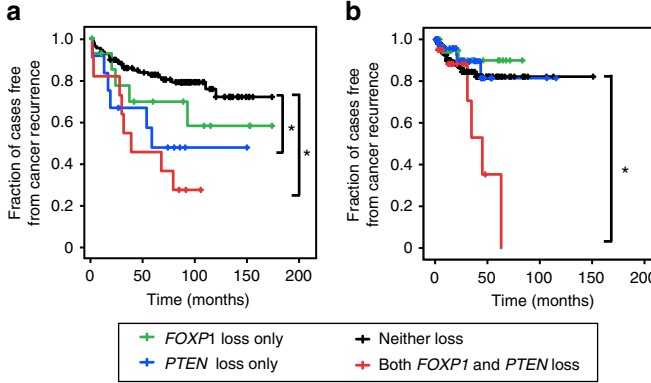

**Fig. 4** Cooperative loss of *FOXP1*, a 3p13-14 deletion gene, and *PTEN* is associated with human prostate cancer recurrence. **a** Kaplan–Meier curve of biochemical recurrence in the MSKCC primary prostate cancer cohort by combined *FOXP1* and *PTEN* loss, *FOXP1* only loss, *PTEN* only loss, or loss of neither. Log-rank *P < 0.05. **b** Kaplan–Meier curve of biochemical recurrence in the TCGA prostate cancer cohort as in **a**. Log-rank *P < 0.05

**Table 1 Combined *PTEN* loss and *SHQ1-FOXP1* loss is associated with recurrence in primary prostate, breast, and head and neck cancer by multivariate Cox regression in TCGA cohorts**

| Cancer cohort | P-value | Hazard ratio | 95% CI for HR | |
|---|---|---|---|---|
| | | | Lower | Upper |
| *Prostate cancer (TCGA cohort)* | | | | |
| Indicator (neither *FOXP1* nor *PTEN* loss) | NS | | | |
| *FOXP1* loss only (expression $z < -1$ or hemizygous or homozygous loss) | NS | 0.607 | 0.179 | 2.061 |
| *PTEN* loss only (expression $z < -2$ or hemizygous or homozygous CN loss) | NS | 0.779 | 0.289 | 2.103 |
| Both *FOXP1* and *PTEN* loss | 0.029 | 2.805 | 1.112 | 7.075 |
| *Breast cancer (TCGA cohort)* | | | | |
| Indicator (neither *FOXP1-SHQ1* nor *PTEN* loss) | NS | | | |
| *FOXP1-SHQ1* loss only (expression $z < -1$ or hemizygous or homozygous loss) | NS | 1.190 | 0.563 | 2.515 |
| *PTEN* loss only (expression $z < -2$ or hemizygous or homozygous CN loss) | NS | 1.562 | 0.858 | 2.843 |
| Both *FOXP1-SHQ1* loss and *PTEN* loss | 0.042 | 2.059 | 1.028 | 4.123 |
| *Head and neck cancer (TCGA cohort)* | | | | |
| Indicator (neither *FOXP1-SHQ1* nor *PTEN* loss) | NS | | | |
| *FOXP1-SHQ1* loss only (expression $z < -1$ or hemizygous or homozygous loss) | NS | 1.970 | 0.977 | 3.974 |
| *PTEN* loss only (expression $z < -2$ or hemizygous or homozygous CN loss) | NS | 1.359 | 0.297 | 6.220 |
| Both *FOXP1-SHQ1* loss and *PTEN* loss | 0.048 | 2.144 | 1.006 | 4.570 |

HR hazard ratio, CI confidence interval

Memorial Sloan Kettering Cancer Center (MSKCC) primary prostate cancer cohort[3], patients with both *PTEN* loss and *FOXP1* loss had significantly worse biochemical recurrence rates than those with no loss, as determined by Cox regression (hazard ratio, HR, 4.56; 95% confidence interval, CI, 2.02, 10.26; $P < 0.001$; Supplementary Table 3, Fig. 4a). *PTEN* loss alone is also associated with biochemical recurrence in this cohort but with a lower HR (3.07; Supplementary Table 3). In the TCGA prostate cancer cohort[4], combined *FOXP1* and *PTEN* loss is also associated with biochemical recurrence, compared to loss of either gene alone or no loss ($P = 0.029$, Cox regression; HR, 2.81; 95% CI, 1.11, 7.08; Table 1, Fig. 4b). Combined loss of *PTEN* and the 3p14 deletion gene *FOXP1* is therefore prognostic for prostate cancer recurrence, highlighting the functional cooperativity of these alterations.

To look at the clinical significance of co-occurring *FOXP1-SHQ1* and *PTEN* loss more broadly across cancer types, we asked whether combined *FOXP1-SHQ1* and *PTEN* loss was associated with cancer recurrence in all TCGA cohorts with co-enriched *FOXP1-SHQ1* and *PTEN* loss. Of the three of these TCGA cohorts with published recurrence data, breast cancer[5] showed a significant association between recurrence and combined *FOXP1-SHQ1* and *PTEN* loss, where loss was defined as copy number or expression loss ($P = 0.042$, Cox regression; HR = 2.06; 95% CI, 1.03, 4.12; Table 1). This is consistent with findings that FOXP1 protein expression loss correlates with poor prognosis and relapse in breast cancer[32]. Of the two additional TCGA cohorts with recurrence data available through the MSKCC cBio portal[18], head and neck cancer[8] showed a significant association between recurrence and combined *FOXP1-SHQ1* and *PTEN* loss ($P = 0.048$, Cox regression; HR = 2.14; 95% CI, 1.01, 4.57; Table 1). These results raise the possibility that co-occurring *FOXP1-SHQ1* and *PTEN* loss may cooperate to impact outcomes across multiple cancer types.

## Discussion

The work presented here establishes that the 3p13-14 locus from *FOXP1* to *SHQ1* has tumor suppressor activity in the context of PI3K pathway activation conferred by *PTEN* loss. In addition to accelerating cancer progression, deletion of the *Foxp1-Shq1* locus

with *Pten* loss results in changes in signaling, such as mTORC1 activation and restored nuclear hormone receptor activity, that are mirrored in human prostate cancers with comparable genotypes. Collectively, this work establishes the oncogenic cooperativity between frequent *FOXP1-SHQ1* loss at 3p13-14 and the central PI3K pathway.

Loss of multiple genes at the 3p13-14 locus may contribute to this cooperativity and its association with cancer outcome. As observed, loss of 3p13-14, including loss of the transcriptional regulator *FOXP1*, modulates PI3K-mediated androgen axis repression. FOXP1 has been observed to repress AR transcriptional activity. Additionally, FOXP1 acts as an estrogen receptor transcriptional co-activator in breast cancer[33], consistent with association of its loss with poorer outcomes. This may extend to endometrial cancers, which are commonly ER-driven. Some subtypes of head and neck cancer show moderate AR positivity, while others have reports of sporadic tamoxifen sensitivity[34]. It is interesting to speculate that functional interactions between *FOXP1* and the nuclear hormone receptors AR and ER[17] may play a role in the 3p13-14 deletion's prognostic significance in prostate and breast cancers and potentially others.

Since we observe co-enrichment of *FOXP1-SHQ1* and *PTEN* deletion in a variety of cancer types, the potential cooperativity of several genes in the *FOXP1-SHQ1* locus with *PTEN* loss could provide multiple mechanisms depending on the cancer type. Beyond nuclear hormone receptor-based synergy, the PI3K pathway has been shown to positively regulate FOXP1 expression and its downstream transcriptional effects[33]; *FOXP1* deletion in the context of *PTEN* loss therefore allows tumors to circumvent the induction of tumor suppressive FOXP1-mediated transcriptional activity, which would otherwise result from *PTEN* loss-mediated PI3K activation. Moreover, the other genes in the deletion, *EIF4E3*, *RYBP*, *PROK2*, and *GPR27* all mediate activities that intersect with the PI3K pathway in a potentially tumor suppressive manner. *RYBP* loss is known to result in ubiquitination and subsequent degradation of p53[35], the loss of which synergizes with *PTEN* loss in prostate cancer[36]. EIF4E3 is known to antagonize EIF4E1[10, 11], and its loss has the potential to remove the brakes on TOR pathway-induced growth through S6K-phosporylated 4E-BP1. Last, we observed both ERK activation and hyperactivation of S6 beyond that caused by PTEN

deletion alone in the tumors with both FOXP1-SHQ1 deletion and PTEN loss. This enhanced signaling may be another mechanism for cooperativity. Notably, PROK2, as a peptide cytokine that activates the prokinectin G-protein-coupled receptors, has been implicated in MEK and ERK1/2 activation[37] as well as phosphoinositide turnover[38]. PROK2 deletion may therefore mediate some level of negative feedback of the ERK activation observed in our transgenic mice and thereby modulate oncogenic stress in tumors with combined PTEN and FOXP1-SHQ1 deletion. Alternatively, since this G-coupled receptor signaling is followed by rapid attenuation of downstream ERK activation through both heterologous and homologous mechanisms[37], PROK2 deletion might attenuate a mechanism of ERK feedback inhibition. Interestingly, GPR27, an orphan G-protein-coupled receptor, has also been shown to mediate phosphoinositol-mediated phosphorylation of ERK and Akt[39]; GPR27 deletion may therefore have similar effects to PROK2 loss on ERK signaling. Future work may explore the contributions of these genes and their interplay in PI3K-driven cancer progression and its molecular underpinnings.

Increasingly, many large genomic alterations appear to contain multiple drivers that act concurrently, as recently demonstrated for 8p and 17p deletions[1, 2]. It will be of particular interest in future work to determine whether the oncogenic effects of FOXP1-SHQ1 deletion are explained by loss of individual genes within the locus or by the collective loss of multiple genes in the region. While the answer will ultimately require selective deletion of genes within the locus alone and in various combinations, several lines of evidence support a multigenic model. First, the loxP sites in Foxp1-Shq1 locus in our model were engineered in a manner that allows conditional deletion of FoxP1 and Shq1 individually or deletion of the entire locus. The fact that all the prostate tumors are enriched for deletion of the entire locus or for deletion of both FoxP1 and Shq1, rather than selective deletion of FoxP1 and/or Shq1 individually, suggests that at least two genes in the interval play a role in restraining tumor progression in this model. Second, the negative prognostic impact of FOXP1-SHQ1 deletion attributable largely to FOXP1 loss suggests FOXP1 may have tumor suppressor activity, consistent with the known suppressive effects of FOXP1 on hormone receptor signaling[16, 17]. This observation implicating FOXP1, together with an earlier report implicating Shq1 loss in a mouse model of Pten$^{-/-}$; Tp53$^{-/-}$ driven metastasis[25], provides additional support for a multigenic model. Third, somatic mutations have been reported across a range of human tumor types in several genes within the locus (FOXP1, RYBP, SHQ1). Although less frequent than the complete FOXP1-SHQ1 deletion, the recurrent nature of these mutations, particularly those in the forkhead domain of FOXP1 and in the dyskerin interaction domain of SHQ1 support a functional role for these genes[4, 18].

The work presented here suggests that multigenic and multi-driver alterations, such as 3p13-14, may functionally cooperate with other major alterations such as PTEN loss to promote cancer. This cooperativity may also occur through genes within multigenic alterations that have distinct functions at different stages of disease initiation and progression. As such multigenic alterations are further explored, the functional–structural reasons that linked cancer drivers exist at single alteration loci may be elucidated and their prognostic potential utilized.

## Methods

**Human cancer cohort analyses.** For all analyses, only primary tumors without neoadjuvant treatment were included. The following published cohorts were used and filtered for only primary, non-neoadjuvantly treated cases: MSKCC prostate adenocarcinoma (2010)[3] ($n = 146$), TCGA prostate adenocarcinoma (2015)[4] ($n = 330$), Cornell/Broad neuroendocrine prostate cancer (only the neuroendocrine subtype cases used in our analysis, $n = 44$)[20], and the 12 published TCGA cancer cohorts that had PTEN and FOXP1-SHQ1 loss in >2% of cases (uterine endometrioid cancer[40], $n = 240$; stomach adenocarcinoma[6], $n = 286$; breast carcinoma[5], $n = 817$; lung adenocarcinoma[41], $n = 230$; lung squamous carcinoma[42], $n = 178$; renal clear cell carcinoma[7], $n = 418$; head and neck squamous carcinoma[8], $n = 279$). Two other published TCGA cohorts, papillary thyroid carcinoma (2014)[43] and acute myeloid leukemia (2013)[44], at present did not have >2% PTEN and FOXP1-SHQ1 loss and were therefore not analyzed for co-occurrence associations. The data meet the assumptions of the tests and have appropriate variance. Analyses were carried out in SPSS and R[45].

To analyze genetic co-enrichments, Fisher's exact test was used to identify significant co-enrichment of copy number loss. PTEN CN loss was defined as hemizygous/shallow or homozygous/deep deletion (mutations can be included in this definition without changing the PTEN loss sample set due to overlap with copy number losses). FOXP1 and SHQ1 individual CN loss was similarly defined as hemizygous/shallow or homozygous/deep deletion. FOXP1-SHQ1 CN loss was defined as hemizygous/shallow or homozygous/deep deletion of all genes from FOXP1 to SHQ1, inclusive. For the TCGA prostate cohort comparison of (a) PTEN CN loss to FOXP1 CN loss, $n = 19$ both lost, $n = 73$ PTEN CN loss only, $n = 30$ FOXP1 CN loss only, $n = 208$ neither lost; (b) PTEN CN loss to SHQ1 CN loss, $n = 20$ both lost, $n = 72$ PTEN CN loss only, $n = 31$ SHQ1 CN loss only, $n = 207$ neither lost; (c) PTEN CN loss to FOXP1-SHQ1 CN loss, $n = 15$ both lost, $n = 77$ PTEN CN loss only, $n = 25$ FOXP1-SHQ1 CN loss only, $n = 213$ neither lost. Fisher's exact test was similarly used to identify significant co-enrichment of copy number loss between PTEN CN and FOXP1-SHQ1 CN loss in the 12 published TCGA cancer cohorts that had PTEN and FOXP1-SHQ1 loss in >2% of cases (listed above).

For survival analyses, Cox regression was used to test the association between cancer recurrence and the following loss groups: FOXP1-SHQ1 loss alone, PTEN loss alone, combined FOXP1-SHQ1 and PTEN loss, and neither lost. The association with cancer recurrence was also tested for the following set of groups: loss of each individual gene in the FOXP1-SHQ1 locus alone, PTEN loss alone, combined loss of the individual gene and PTEN, and neither lost. For the prostate cancer survival analyses, the most current published biochemical recurrence was used[6, 23]. For the MSKCC prostate cancer cohort[7], FOXP1 loss and other individual gene loss within the FOXP1-SHQ1 locus was defined as hemizygous/shallow or homozygous/deep deletion or RNA expression loss at $z < -1$; FOXP1-SHQ1 locus loss was defined as hemizygous/shallow or homozygous/deep deletion or RNA expression loss at $z < -1$ of all genes in the locus; PTEN loss was defined as hemizygous/shallow or homozygous/deep deletion or expression loss at $z < -2$. For the TCGA prostate cancer cohort[6], FOXP1 loss and other individual gene loss within the FOXP1-SHQ1 locus was defined homozygous/deep deletion or RNA expression loss at $z < -1$; FOXP1-SHQ1 locus was defined as hemizygous/shallow or homozygous/deep deletion or RNA expression loss at $z < -1$; PTEN loss was defined as hemizygous/shallow or homozygous/deep deletion or expression loss at $z < -2$. For the pan-cancer survival analyses, three published TCGA cohorts that showed enriched co-occurrence of FOXP1-SHQ1 and PTEN loss also had published recurrence data available: TCGA breast invasive carcinoma (2015)[9], TCGA uterine endometrioid (2013)[15], and TCGA stomach adenocarcinoma (2014)[10]. These were analyzed by Cox regression for associations between recurrence and the loss categories described above. FOXP1 loss and other individual gene loss within the FOXP1-SHQ1 locus was defined as hemizygous/shallow or homozygous/deep deletion or RNA expression loss at $z < -1$; FOXP1-SHQ1 locus loss was defined as hemizygous/shallow or homozygous/deep deletion or RNA expression loss at $z < -1$ of all genes in the locus; PTEN loss was defined as hemizygous/shallow or homozygous/deep deletion or expression loss at $z < -2$.

**Transgenic mouse generation and analysis.** Foxp1-Shq1$^{flox}$ conditional knockout transgenic mice (M. musculus) were generated by targeting the SHQ1 conditional frameshift construct HTGRS0100_A_A12 (EUCOMM, KO first allele reporter-tagged insertion with conditional potential) into Foxp1$^{f/f}$ ES cells generated from the Foxp1$^{f/f}$ conditional knockout mice in Feng et al.[46], which were then injected into C57BL/6 blastocysts. SHQ1$^{flox}$ conditional knockout transgenic mice were generated by targeting the same SHQ1 conditional frameshift construct HTGRS0100_A_A12 (EUCOMM) into 129 ES cells and injected into C57BL/6 blastocysts by the Rockefeller University Gene Targeting Facility. After generation of the targeted transgenic mouse lines, they were converted from a reporter-tagged knockout to a conditional knockout by breeding with a FLPase line (B6.Cg-Tg (ACTFLPe)9205Dym/J, Jackson Labs) and selection for loss of the neo cassette by PCR. FOXP1-SHQ1$^{flox}$ and SHQ1$^{flox}$ mice were bred with Pb-Cre4 mice (Jackson Labs strain 01XF5) and Pten$^{flox/flox}$ mice[30, 47]. Male mice were analyzed at 6, 9, and 12 months of age. All animal work was approved by the MSKCC Institutional Animal Care and Use Committee.

For genomic DNA PCR analysis of the transgenic mouse tumors, DNA was extracted from FFPE slides using the AllPrep DNA/RNA FFPE Kit (Qiagen). Primers amplifying the floxed or recombined allele of FOXP1 (AGAAGATCCGTTGACCTGCA, GAACACTGTCGAATGACCCTGC, and ACGTGCCCATTTCTTCAGGT), wild-type, floxed, or recombined SHQ1 (CACCTGTGTTGCTAACGTTCCTTC, CTACTGTGGCTACTTCAAGGATTACC,

and GGGTTTCTTCACTGCTCGAG), and the recombined allele of the entire *FOXP1-SHQ1* locus (AGAAGATCCGTTGACCTGCA, GCATAACGATACCAC GATATCAA) were used for 38 cycles of PCR. For quantitative RT-PCR, RNA isolated from frozen tissue (1.5 µg) was converted to cDNA using the High-Capacity RNA-to-cDNA kit (ThermoFisher), and analyzed by real-time PCR using 2× QuantiFast SYBR Green PCR Master Mix (Qiagen). Primers specific for the full-length unrecombined alleles of *FOXP1* (AGAACGCGGAAGTTAGACCA and GCTGCTTTTCTGGAGATTC) and *SHQ1* (GTACTTCGAGGGGGTGGACT and CATAGGTCCCCTGCTCAGAT) were used, and the Ct values were normalized to *GAPDH* (TGCACCACCAACTGCTTAGC and GGCATGGACTGTGGT CATGAG). Three technical replicates were carried out per sample and at least three biological samples per experimental condition (per genotype per age) were analyzed; the PCR experiments were replicated twice.

**Transgenic mouse tumor histology and characterization**. Mouse prostate dissection was performed and prostates were fixed in 4% paraformaldehyde for 18 h and embedded in paraffin.

After sectioning, the immunohistochemical detection of AR, Ki67, pS6, pAkt, pERK, and PTEN was performed at the Molecular Cytology Core Facility of MSKCC using the Discovery XT processor (Ventana Medical Systems). The tissue sections were deparaffinized with EZPrep buffer (Ventana Medical Systems), antigen retrieval was performed with CC1 buffer (Ventana Medical Systems) and sections were blocked for 30 min with Background Buster solution (Innovex), followed by avidin-biotin blocking (Ventana Medical Systems) for 8 min. Anti-AR (Epitomics cat #3184-1, 0.66 µg/ml), anti-Ki67 (Vector, cat #VP-K451, 0.4 µg/ml), anti-pS6 (Cell Signaling, cat #2211L, 0.12 µg/ml), anti-PTEN (Cell Signaling cat #9188, 4 µg/ml), anti-pErk1/2 (Cell Signaling, cat #4370, 1 µg/ml), and anti-pAkt Ser473 (Cell Signaling, cat #4060, 1 µg/ml) antibodies were applied and sections were incubated for 5 h, followed by 60 min incubation with biotinylated goat anti-rabbit IgG (Vector Labs, cat #PK6101) at 1:200 dilution. For anti-pAkt antibodies, TSA amplification was performed by incubation with of Streptavidin-HRP (Ventana) for 12 min followed by incubation with TSA (PerkinElmer, cat #FP1019, 1:100 in amplification buffer) for 16 min. Detection was performed with DAB detection kit (Ventana Medical Systems). Slides were counterstained with hematoxylin.

Immunohistochemical detection of cytokeratin 8/18, cytokeratin 5, and cytokeratin 14 were carried out by the MSKCC-Rockefeller-Weill Cornell Laboratory of Comparative Pathology. For CK8/18, paraffin sections were dewaxed in xylene and hydrated into graded alcohols. Endogenous peroxidase activity was blocked by immersing the slides in 1% hydrogen peroxide in PBS for 15 min. Pretreatment was performed in a steamer using 10 mM citrate buffer, pH 6.0 for 30 min. Sections were incubated overnight with primary cytokeratin 8/18 antibody (Anti-Guinea Pig antibody from Fitzgerald, Cat. No. RDI-PROGP11) diluted 1:1000. Sections were washed with PBS and incubated with the appropriate secondary antibody followed by avidin-biotin complexes (Vector Laboratories, Burlingame, CA, Cat. No. PK-6100). Antibody reaction was visualized with 3-3′ diaminobenzidine (Sigma, Cat. No. D8001) and counterstained with hematoxylin. Tissue sections were dehydrated in graded alcohols, cleared in xylene, and mounted. Cytokeratin 5 (Covance, Cat. PRB-160P diluted 1:1000) and cytokeratin 14 (Thermo, Cat. RB-9020, diluted at 1:750) IHC was performed on a Leica Bond™ RX using the Bond™ Polymer Refine Detection Kit (Cat. No. DS9800). The sections stained for CK5 were pre-treated using heat-mediated antigen retrieval with Citrate, pH 6 (Leica Biosystem Epitope Retrieval 1, Cat. No. AR9961) for 20 min. Sections stained for CK14 were pre-treated using heat-mediated antigen retrieval with EDTA pH 9 (Leica Biosystem Epitope Retrieval Solution 2, Cat. No. AR9640) for 20 min. DAB was used as the chromogen, counterstained with hematoxylin, and mounted. At least four biological replicates (mice) per experimental condition (per genotype per age) were analyzed.

For western blot analysis, tissue underwent bead-based lysis (Lysing Matrix A, MP Biomedicals) in RIPA buffer; after BCA-based protein quantitation (Pierce) and normalization, samples were then run on a 4–12% Bis-Tris gel (NuPAGE, Thermofisher), transferred to nitrocellulose membrane (Immobilon IPVH00010) and blocked (TBST with 5% albumin). Antibodies used were against AR (AR CST, cat #5153; 1:1000) and actin (CST, cat #4970, 1:10,000).

For Ki67 quantitation, percent Ki67 positivity was quantified by counting Ki67-positive and total number of cells in three 40× fields per mouse. At least four biological replicates (mice) per experimental condition (per genotype per age) were analyzed.

Mouse pathology was scored by a board-certified pathologist (A.G.) who was blinded to the genotype. The phenotypes[24] were scored as follows: (a) wild-type histology: lack of or only mild increase in number of epithelial cells with architecture ranging from single cell thick epithelial lining to mild tufting and papillary changes and absence of significant nuclear atypia; (b) high-grade PIN: moderate proliferation of atypical epithelial cells within glands, without invasion through the basement membrane was scored; architecture ranges from tufting to papillary to cribriform with moderately increased nuclear atypia; (c) intraductal carcinoma: marked proliferation of epithelial cells within glands; atypical cells with severe nuclear atypia fill almost the entire duct lumen with bulging into the stroma, but no frank invasion; (d) invasive carcinoma: invasion of the tumor cells into adjacent stroma as single cells, irregular nests, clusters or sheets.

**RNA-seq**. Frozen mouse prostate tissue (pooled lobes) was submerged in Trizol and homogenized using a FastPrep-24 instrument with Lysing Matrix A (MP Biomedicals). After chloroform extraction using Phase-lock Heavy spin tubes for phase separation, isopropanol precipitation was carried out. The RNA was further purified using the RNEasy Mini kit (Qiagen).

After ribogreen quantification and quality control by Agilent BioAnalyzer, 500 ng of total underwent polyA selection and Truseq library preparation according to instruction provided by Illumina (TruSeq™ RNA Sample Prep Kit v2), with six cycles of PCR. Samples were barcoded and run on a Hiseq 2500 in a single-read 50 bp run, using the TruSeq SBS Kit v3 (Illumina). On average, 49.5 million reads were generated per sample. At the most the ribosomal reads represented 0.3% and the percent of mRNA bases was closed to 70% on average. Three biological samples per experimental condition (per genotype per age) were analyzed.

FASTQ output was mapped to the genome (UCSC MM10, Mus_musculus. GRCm38.80) using the rnaStar aligner[48] that maps reads genomically and resolves reads across splice junctions. The two pass mapping method outlined in Engström et al.[49] in which the reads are mapped twice was then used. The first mapping pass uses a list of known annotated junctions from Ensemble. Novel junctions found in the first pass are then added to the known junctions and a second mapping pass is done (on the second pass the RemoveNoncanonical flag is used). After mapping, the output SAM files were post processed using PICARD and converted into compressed BAM format. Expression counts were derived from the mapped reads using HTSeq. DESeq2 was then used to normalize the data set and analyze differential expression. Heat maps were generated using the rlog-transformed expression of the top 1000 significantly altered genes with Euclidean distance and complete agglomeration for hierarchical clustering. Analyses were carried out in R. For GSEA, differentially expressed genes (defined as genes differentially expressed between *FOXP1-SHQ1*[f/f];*PTEN*[f/f] 12mo murine prostate tumors relative to *PTEN*[f/f] 12mo murine prostate tumor with $P < 0.05$, Wald) were ranked by fold change and analyzed by GSEA (GSEAPreranked, classic scoring scheme, Broad GenePattern, using MSigDB gene sets c2–c7). The RNA-seq data are deposited under Gene Expression Omnibus (GEO) series GSE83358.

**Androgen-regulated gene expression analysis**. The differences in androgen signaling in the transgenic mice were examined using an androgen-regulated signature derived in LNCaP cells and human prostate cancer[50], and, a second, mouse castration-derived androgen-induced signature[30]. The significance of the differences in these signatures in the RNA-seq data from *Foxp1-Shq1*[flox/flox], *Pten*[flox/flox] and *Pten*[flox/flox] mouse prostate at 12 months was tested by t-test. The differences in androgen signaling in human primary prostate cancer were tested (TCGA prostate cohort[4], $n = 330$ total, $n = 72$ *PTEN* loss only, $n = 49$ *FOXP1* loss only, $n = 22$ both *PTEN* and *FOXP1* loss, $n = 187$ neither lost) by ANOVA with Tukey post hoc correction (all pairwise comparisons tested) using same androgen-regulated signature[4, 50] used for the mouse data, but as reported for the TCGA prostate cohort[3].

For this analysis, *PTEN* loss was defined as copy number loss (hemizygous or homozygous) or expression loss ($z < -2$). *FOXP1* loss was defined as copy number loss (homozygous) or expression loss ($z < -1$). ANOVA with Tukey post hoc correction was used to assess differences between loss categories.

**Data availability**. The RNA-seq data set generated during the current study are available in the GEO repository under GEO Series accession GSE83358, http://www.ncbi.nlm.nih.gov/geo/query/acc.cgi?acc = GSE83358.

The published human data sets analyzed herein are available, including copy number calls and outcome data, as follows: MSKCC prostate adenocarcinoma (GSE21032, http://www.cbioportal.org/study?id = prad_mskcc)[3], TCGA prostate adenocarcinoma (http://www.cbioportal.org/study?id = prad_tcga_pub)[4], Cornell/ Broad neuroendocrine prostate cancer (dbGap phs000909.v.p1, http://www. cbioportal.org/study?id = nepc_wcm_2016)[20], TCGA breast invasive carcinoma (http://www.cbioportal.org/study?id = brca_tcga_pub2015)[5], TCGA head and neck squamous carcinoma (http://www.cbioportal.org/study?id = hnsc_tcga_pub)[8], TCGA uterine endometrioid cancer (http://www.cbioportal.org/study?id = ucec_tcga_pub)[40], TCGA stomach adenocarcinoma (http://www.cbioportal.org/ study?id = stad_tcga_pub)[6], TCGA lung adenocarcinoma (http://www.cbioportal. org/study?id = luad_tcga_pub)[41], TCGA lung squamous carcinoma (http://www. cbioportal.org/study?id = lusc_tcga_pub)[42], TCGA renal clear cell carcinoma (http://www.cbioportal.org/study?id = kirc_tcga_pub)[7], TCGA colorectal adenocarcinoma (http://www.cbioportal.org/study?id = coadread_tcga_pub)[51], TCGA kidney chromophobe cancer (http://www.cbioportal.org/study?id = kich_tcga_pub)[52], TCGA glioblastoma (http://www.cbioportal.org/study?id = gbm_tcga_pub2013)[53], TCGA bladder urothelial carcinoma (http://www. cbioportal.org/study?id = blca_tcga_pub)[54], and TCGA ovarian serous carcinoma (http://www.cbioportal.org/study?id = ov_tcga_pub)[55]. All data for the TCGA cohorts are also available from https://cancergenome.nih.gov/. All other remaining data are available within the Article and Supplementary Files, or available from the authors upon request.

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

## Acknowledgements

This work has been supported by Howard Hughes Medical Institute and NIH grants CA155169, CA19387, CA092629, and CA008748 (to C.L.S.) and R01 CA182503-01A1 and P50 CA092629-15 (to B.S.C.). We thank the MSKCC Molecular Cytology Core, the MSKCC-Rockefeller-Weill Cornell Laboratory of Comparative Pathology, and the MSKCC Integrated Genomics Operation Core for technical work. The MSKCC Molecular Cytology Core is supported by CA008748. The MSKCC Integrated Genomics Operation Core is funded by P30 CA08748, Cycle for Survival and the Marie-Josée and Henry R. Kravis Center for Molecular Oncology.

## Author contributions

H.H., J.W., B.S.C., N.M. and C.L.S. designed, created, and characterized the transgenic animals. H.H., B.S.C., R.M. and A.G. performed the histopathological analysis. P.J.I. performed the genomic PCR experiments. H.H. performed and analyzed the RNA-

sequencing experiments and carried out the human cohort analyses. H.H. and C.L.S. wrote the initial draft of the manuscript and all authors contributed to the final version.

## Additional information

**Competing interests:** The authors declare no competing financial interests.

