## [Peer review file · Nature Communications]

Reviewers' comments:

Reviewer #1 (Remarks to the Author):

This is an interesting study that demonstrates and aims to investigate the functional relevance of the co-deletion of FOXP1-SQ1 (3p13-14) and PTEN in prostate and other cancers. The authors show that Pten and Foxp1-Shq1 cooperate in prostate cancer progression, and they extend the findings to analyses of compound mouse models, which is impressive. The paper may benefit from the suggestions below:

1. Fig.2: Can the authors show a survival (incidence) curve to show the acceleration of the phenotype in the mice?

2. Fig.2b: How exactly was the pathology scored? Not described the text or methods.

Related to the point above, the histology says that the lobes were pooled (?) Is this a mistake. (p13 -lines 259-260).

3. Fig.2d: Any reason that GPR27 levels are reduced so much more than the other genes?

3. Fig.3a and b: I do not quite understand the absence of AR expression in the compound mice but up-regulation of AR genes?

4. Also related the above, can the authors show the AR in a more standard way like a heat map? (Some data are in the supplement)

5. The signaling pathway analyses would benefit from western blot analyses.

6. Considering Fig.1, can the authors extend the generality? Do the genes cooperate for outcome in other cancers (Fig.4)? If so, would they expect the mechanism to be different than affecting AR signaling? (Which would be relatively restricted to prostate). Here the authors can possibly show synergy of association with outcome in other cancers and to discuss alternative mechanisms. (now cited as supp data I think it should be in main body).

Reviewer #2 (Remarks to the Author):

Hieronimus et al. present a manuscript of their thorough and elegant study named "Deletion of 13p13-14 locus spanning FOXP1 to SHQ1 cooperates with PTEN loss in prostate oncogenesis".

Although the locus at 3p13-14 is commonly deleted in prostate cancer, and the Foxp1-Shq1 deletion in mice has been associated with metastasis increase, this is the first time when a multigenic deletion FOXP1-SHQ1 is shown to be a driver event with prognostic value.

Questions/comments:

In the results, PIK3CB mutation and amplification is said to be significantly enriched in tumors with FOXP1-SHQ1 deletion, and Supplementary Table 1 is given here as the reference. However, there's nothing on PIK3 there, neither on point mutations when/if included as PTEN alterations. Moreover, it should be opened why PIK3 (PIK3A/3B) is of importance here (PI3K/AKT/mTOR pathway, also lines 117 and 197).

Do deletions of GPR27 and PROK2 have any affect?

Please add numbers of cohort samples to the main methods section for the MSKCC and TCGA

cohorts. Now they are only found in Figure 1.

Figure 1: Why is GPR27 not shown? The colors in the bar graph need to be explained. Now it's confusing as one tries to look explanation from the copy number scale bar.

Figure 2: Histogram picture is hard to understand as the numbers are mentioned only in the text but not shown in the picture. Please add to the picture. Instead of SEM, SD should be used. SEM is not a descriptive statistics.

Figure 3: Explain colors in the bars in b). Again, instead of SEM, SD should be used. For the statistical comparisons, it is not clear which groups were tested in d). What about "no loss" vs "FOXP1-SHQ1 loss"?

Supplementary Figure 3: Upside down in its present form. In the figure text 16 should be 14 (as for CK14).

Reviewer #1:

This is an interesting study that demonstrates and aims to investigate the functional relevance of the co-deletion of FOXP1-SQ1 (3p13-14) and PTEN in prostate and other cancers. The authors show that Pten and Foxp1-Shq1 cooperate in prostate cancer progression, and they extend the findings to analyses of compound mouse models, which is impressive. The paper may benefit from the suggestions below:

1. Fig.2: Can the authors show a survival (incidence) curve to show the acceleration of the phenotype in the mice?

Yes, we have now added cumulative incidence survival curves of intraductal carcinoma and invasive carcinoma as Supplementary Fig. 2 and added it to the text (p. 5, paragraph 2). Since the mice were analyzed at 6, 9, and 12 month time points only, the incidence times reflect this.

2. Fig.2b: How exactly was the pathology scored? Not described the text or methods.

We thank the reviewers for pointing out this omission. We have added a description of the blinded pathology scoring, including how we scored murine high grade PIN, intraductal carcinoma, and invasive carcinoma, to the methods (p. 13, paragraph 3, Methods, "Pathology") and added references to Ittmann *et al.*, *Cancer Res*, 2013 (Animal models of human prostate cancer: the consensus report of the New York meeting of the Mouse Models of Human Cancers Consortium Prostate Pathology Committee) to the text and methods (p.6, paragraph 1; Methods, p. 13, paragraph 3).

Related to the point above, the histology says that the lobes were pooled (?) Is this a mistake.(p13 - lines 259-260).

The lobes were pooled for the RNAseq analysis but not for the histology. Thank you for pointing this out. We have corrected this in the Methods (p. 12, paragraph 3; p. 13, paragraph 4).

3. Fig.2d: Any reason that GPR27 levels are reduced so much more than the other genes?

We believe this question refers to the decrease in *Prok2* RNA level, which shows a greater reduction than the other genes (Figure 2d, directly above the label for GPR27, which shows less of a reduction than the other genes), rather than *Gpr27*. We've increased the separation between the GPR27 label and the PROK2 graph to eliminate this confusion in the future.

To address the question, *Prok2* exhibits positive feedback activation of its own transcription (Cheng *et al.*, *Nature*, 2002) and therefore heterogenous deletion of this secreted cytokine may result in greater reduction in *Prok2* levels in adjacent cells which have incomplete deletion. Moreover, the observed PROK2 RNA levels were low in both genetic backgrounds, so the relative decrease is more sensitive to

experimental noise. We have added commentary about these potential causes of the greater *Prok2* reduction in the text (p. 5, paragraph 2).

3. Fig.3a and b: I do not quite understand the absence of AR expression in the compound mice but up-regulation of AR genes?

We thank the reviewer for the opportunity to clarify this issue. AR is expressed, albeit at a lower level, in the compound mice, as indicated by the unchanged level of AR RNA and the light AR staining by IHC. We have added a Western (Supplementary Fig. 6d) showing the AR expression in these mouse prostate tumors.

Regarding the up-regulation of AR target genes in the context of lower AR levels, FOXPI is an AR transcriptional co-repressor, and therefore loss of FOXPI may de-repress the existing AR. This would effectively make the pool of active or non-repressed AR larger and restore AR target gene expression. Future work in organoid culture will test these hypotheses. We have added a longer commentary on these possibilities (p. 7, paragraph 2).

4. Also related the above, can the authors show the AR in a more standard way like a heat map?

We have added heat maps to Figure 3 to show the AR responsive gene expression levels (AR signatures, Fig. 3c and 3d), as well as a graph of AR RNA level (Supplemental Fig. 6). We have also retained the AR signature score representation of the AR signatures (Figure 3, Supplemental Fig. 6) as this has become one common way of summarizing AR responsive gene effects in a compact quantitative manner, as in the TCGA prostate cohort publication (Cancer Genome Atlas Research Network, *Cell*, 2015).

5. The signaling pathway analyses would benefit from western blot analyses.

The tumors in this model have a heterogeneous mix of tumor cells and infiltrating stroma, which complicates the interpretation of tumor lysates by western blot. For example, we see residual Pten protein expression in tumor lysates despite lack of Pten expression in tumor cells by IHC. Therefore, we believe IHC provides the best representation of the signaling changes in the tumors, which have the added benefit of allowing correlation with the changes in histology. In future work we plan to address this question using organoid culture (as above for query #3) and by tumor microdissection. This is not currently possible with currently remaining banked tumor tissue.

6. Considering Fig.1, can the authors extend the generality? Do the genes cooperate for outcome in other cancers (Fig.4)? If so, would they expect the mechanism to be different than affecting AR signaling? (Which would be relatively restricted to prostate). Here the authors can possibly show synergy of association with outcome in other cancers and to discuss alternative mechanisms. (now cited as supp data I think it should be in main body).

Yes, the cooperative deletion of *SHQ1-FOXPI* locus and *PTEN* is associated with recurrence in breast cancer and head and neck cancer. We originally presented the prognostic breast cancer association in Supplemental Table 1. As suggested, we moved these results to the main body as Table 1 (p. 24).

To address the question of additional cancer types, we expanded our analysis to looked at the cancer types which also show co-enrichment of *SHQ1-FOXPI* and *PTEN* deletion but with outcome data available

through the MSK cbio portal (head and neck, stomach). We find that co-deletion is also associated with recurrence in head and neck cancer. We now include this in Table 1 (p. 24).

Additionally, we extend our discussion of the mechanisms by which cooperative *FOXP1-SHQ1/PTEN* deletion may mediate the observed association with outcome (p. 9, paragraphs 2 and 3). Since we observe co-enrichment of *FOXP1-SHQ1* and *PTEN* deletion in a variety of cancer types, the potential cooperativity of several genes in the *FOXP1-SHQ1* locus with *PTEN* loss could provide multiple mechanisms that play roles to varying degrees depending on the cancer type. Beyond nuclear hormone receptor-based synergy, the PI3K pathway has been shown to positively regulate FOXP1 expression and its downstream transcriptional effects. Therefore, *FOXP1* deletion in the context of *PTEN* loss allows tumors to circumvent the induction of tumor suppressive FOXP1 activity which would otherwise result from *PTEN* loss-mediated PI3K activation. Moreover, the other genes in the deletion, *EIF4E3*, *RYBP*, *PROK2*, and *GPR27* all mediate activities that intersect with the PI3K pathway in a potentially tumor suppressive manner, as we expand upon in the discussion (p. 9, paragraph 3). Last, we observed both ERK activation and hyperactivation of S6 beyond that caused by *PTEN* deletion alone in the tumors with both *FOXP1-SHQ1* deletion and *PTEN* loss. This synergistic signaling in known oncogenic pathways may be another mechanism for cooperativity in a range of cancer types.

Reviewer #2:

Hieronymus et al. present a manuscript of their thorough and elegant study named "Deletion of 13p13-14 locus spanning FOXP1 to SHQ1 cooperates with PTEN loss in prostate oncogenesis".

Although the locus at 3p13-14 is commonly deleted in prostate cancer, and the Foxp1-Shq1 deletion in mice has been associated with metastasis increase, this is the first time when a multigenic deletion FOXP1-SHQ1 is shown to be a driver event with prognostic value.

Questions/comments:

In the results, PIK3CB mutation and amplification is said to be significantly enriched in tumors with FOXP1-SHQ1 deletion, and Supplementary Table 1 is given here as the reference. However, there's nothing on PIK3 there, neither on point mutations when/if included as PTEN alterations. Moreover, it should be opened why PIK3 (PIK3A/3B) is of importance here (PI3K/AKT/mTOR pathway, also lines 117 and 197).

We thank the reviewer for the opportunity to expand on this finding and have corrected Supplementary Table 1 to list this association between *FOXP1-SHQ1* deletion and *PIK3CB* amplification/mutation (the specific point mutations now listed). Interestingly, the association between *PI3KB* mutation and *FOXP1-SHQ1* loss primarily occurs in the context of *PTEN* loss ($P = 0.004$ in the subset with *PTEN* loss, NS in the subset without *PTEN* loss). Since *PTEN* deleted tumors are likely dependent on *PIK3CB* due to feedback inhibition of *PIK3CA* activity (Cancer Genome Atlas Research Network, *Cell*, 2015), activating *PIK3CB* mutation may be a mechanism of further boosting the effect of *PTEN* loss, as is the case with the observed E552K mutation (Zhao et al., *Proc Natl Acad Sci U S A*, 2005). We have added this result and comment to the results section (p. 4, paragraph 1).

Do deletions of GPR27 and PROK2 have any affect?

We appreciate the reviewer's interest in *GPR27* and *PROK2* and their interaction with *PTEN* deletion. We have added discussion of *GPR27* and *PROK2* deletion and their possible functional interactions with *PTEN* loss, especially as a G-protein coupled receptor and GPCR ligand respectively that both have effects on ERK and AKT signaling (p. 9, paragraph 3, and p 10, paragraph 1). Analysis of their individual contributions in mouse models is outside the scope of this initial manuscript, given its focus on the effect of the *FOXP1-SHQ1* deletion as a whole. Nonetheless, it is known that germline *PROK2* loss-of-function, as occurs in hypogonadal Kallman syndrome, does not alone cause increased cancer risk in humans or mice (Bianco and Kaiser, *Nat Rev Endocrinol*, 2009), though prostate cancer risk may be affected by the resulting GnRH suppression. We have added further discussion of these genes' deletion to the paper (p. 9, paragraph 3, and p 10, paragraph 1).

Please add numbers of cohort samples to the main methods section for the MSKCC and TCGA cohorts. Now they are only found in Figure 1.

We have added the number of cases in these cohorts to the main methods section (p. 11, paragraph 1).

Figure 1: Why is *GPR27* not shown? The colors in the bar graph need to be explained. Now it's confusing as one tries to look explanation from the copy number scale bar.

We have added *GPR27*, which was not originally included because it is only a provisional RefSeq gene, to Figure 1 as requested.

Thank you for the opportunity to clarify the bar graphs in Figure 1. We have changed the color of the altered samples in the bar graphs to yellow, so they are not confused with the colors in the copy number/heat map scale bar. We have also added a legend for the colors in the bar graphs and updated the legend with a clearer explanation.

Figure 2: Histogram picture is hard to understand as the numbers are mentioned only in the text but not shown in the picture. Please add to the picture. Instead of SEM, SD should be used. SEM is not a descriptive statistics.

We thank the reviewer for suggesting these improvements. We have added the number of mice (per genotype per time point for each disease phenotype) to the histogram (Figure 2b). We have also changed the figure to use SD.

Figure 3: Explain colors in the bars in b). Again, instead of SEM, SD should be used. For the statistical comparisons, it is not clear which groups were tested in d). What about "no loss" vs "FOXP1-SHQ1 loss"?

We have added text to the Figure 3 legend to clarify that the bar colors correspond to different genotypes, which are also listed below each bar, and specify the color for each genotype. We have also changed the figure to use SD. For the statistical comparisons, all pairs were tested (with post-hoc correction), and we have added a description of this to the legend and Methods (p. 13, last paragraph). The "no loss" and "FOXP1-SHQ1 loss alone" groups were compared; they were found not to differ significantly, therefore they don't have an association indicated in Figure 3. We have added this negative finding to the Results text (p.7, paragraph 2).

Supplementary Figure 3: Upside down in its present form. In the figure text 16 should be 14 (as for CK14).

Thank you, we have corrected this (now Supplementary Fig. 4 due to addition of an upstream supplementary figure).

REVIEWERS' COMMENTS:

Reviewer #1 (Remarks to the Author):

The authors have addressed my comments satisfactorily - I have no further comments.

Reviewer #2 (Remarks to the Author):

This revised version of the manuscript includes point-by-point responses to the questions raised. In this new version, authors have added methodological information and have amended the manuscript accordingly. I think this manuscript should be accepted.